# DIRECT EMBEDDING OF TEMPORAL NETWORK EDGES VIA TIME-DECAYED LINE GRAPHS

**Sudhanshu Chanpuriya[1], Ryan A. Rossi[2], Sungchul Kim[2], Tong Yu[2],**
**Jane Hoffswell[2], Nedim Lipka[2], Shunan Guo[2], and Cameron Musco[1]**

[1]University of Massachusetts Amherst, `{schanpuriya,cmusco}@cs.umass.edu`
[2]Adobe Research, `{ryrossi,sukim,tyu,jhoffs,lipka,sguo}@adobe.com`

## ABSTRACT

Temporal networks model a variety of important phenomena involving timed interactions between entities. Existing methods for machine learning on temporal networks generally exhibit at least one of two limitations. First, many methods assume time to be discretized, so if the time data is continuous, the user must determine the discretization and discard precise time information. Second, edge representations can only be calculated indirectly from the nodes, which may be suboptimal for tasks like edge classification. We present a simple method that avoids both shortcomings: construct the *line graph* of the network, which includes a node for each interaction, and weigh the edges of this graph based on the difference in time between interactions. From this derived graph, edge representations for the original network can be computed with efficient classical methods. The simplicity of this approach facilitates explicit theoretical analysis: we can constructively show the effectiveness of our method's representations for a natural synthetic model of temporal networks. Empirical results on real-world networks demonstrate our method's efficacy and efficiency on both link classification and prediction.

## 1 INTRODUCTION

Temporal networks, which are graphs augmented with a time value for each edge, model a variety of important phenomena involving timed interactions between entities, including financial transactions, flights, and web browsing. Common tasks for machine learning on temporal networks include classification of the temporal edges, as well as temporal link prediction, which involves predicting future links given some links in the past. These tasks yield various applications, such as recommendation systems (Zhou et al., 2021) and detection of illicit financial transactions (Pareja et al., 2020). As with most machine learning for graphs, the key to learning for temporal networks is creating effective vector representations, also called embeddings, for the network's components, namely the nodes and edges. These embeddings can either be made as part of an end-to-end framework, or created then passed to off-the-shelf classifiers for downstream tasks; for example, for the edge classification task, a logistic regression classifier can be trained using the training edges' embedding vectors and class labels, then applied at inference time on the test edges' vectors.

The node embedding task, in particular, has seen great interest, and many methods for 'static' (i.e., non-temporal) networks have been proposed over the years (Belkin & Niyogi, 2001; Perozzi et al., 2014; Grover & Leskovec, 2016). Edge embedding has seen less interest; there are some exceptions (Li et al., 2017b; Bandyopadhyay et al., 2019), but generally, edge embeddings are created by first making node embeddings, then aggregating them, e.g., by taking the entrywise product of the two endpoint nodes' embeddings. There are a wide variety of methods for temporal network embedding, including ones based on matrix/tensor factorization (Dunlavy et al., 2011; Li et al., 2017a; Zhang et al., 2018), random walks (Yu et al., 2018; Nguyen et al., 2018), graph convolutional networks (Pareja et al., 2020), and deep autoencoders (Goyal et al., 2018; Rahman et al., 2018; Goyal et al., 2020), but generally, these methods can be seen as variants of those for static networks.

These prior methods overall present some limitations, which we now summarize. With some exceptions, especially in more recent work (Nguyen et al., 2018; Trivedi et al., 2019; Rossi et al.,

|  | NODE REPRESENTATION-BASED | EDGE REPRESENTATION-BASED |
|---|---|---|
| DISCRETE TIME | TIMERS (Zhang et al., 2018) EvolveGCN (Pareja et al., 2020) . . . | DyLink2Vec (Rahman et al., 2018) |
| CONTINUOUS TIME | CTDNE (Nguyen et al., 2018) DyRep (Trivedi et al., 2019) . . . | **This work** |

Table 1: Problem studied in this work.

2020; Wang et al., 2021), these methods often do not work directly with the continuous-valued times of temporal edges, but rather assume that the times are discretized, yielding a sequence of static graphs. Since most datasets have continuous-timed edges, this assumption requires the user to manually determine the discretization and discard precise time information. Second, prior methods generally return embeddings of nodes rather than edges, so edge embeddings can only be calculated indirectly from the nodes, which may be suboptimal for tasks like edge classification. Given that timestamps are associated with edges rather than nodes, and that there are few public datasets where the nodes rather than edges are associated with a time-series of attributes or classes, it is intuitively more natural to derive edge embeddings directly.

We present a simple but novel framework to address these issues: construct the line graph of the network, which converts each edge (timed interaction) of the network into a node, and connects interactions that share an endpoint node. Then, set the edge weights in this line graph based on differences in time between interactions, with interactions that occur closer together in time being connected with higher weights. From this derived graph, which directly represents *topological proximity* (i.e., adjacency of edges) and *temporal proximity*, temporal edge representations for the original network can be computed and exploited with efficient classical methods. To our knowledge, ours is the first method that directly forms embeddings for continuous-time temporal edges without supervision and without aggregating node embeddings - see Table 1. Our method is significantly simpler than recent prior work, particularly compared to deep methods, allowing for more direct theoretical analysis: we propose the union of Gaussian-timed stochastic block models (UGT-SBM), which naturally extends the well-known stochastic block model (SBM) for static networks to temporal networks, and we show that our method can exploit time and community information in UGT-SBMs to form effective representations. Practically, our method's simplicity makes it easy to implement, yet it is accurate and efficient: in experiments on five benchmark real-world temporal networks, our approach achieves superior predictive and runtime performance relative to prior methods.

## 2 METHODOLOGY

Our approach starts with a sequence of timestamped edges:

**Definition 1** (Temporal Graph). *A temporal graph $G = (V, E)$ is a set of vertices $V$ and a set of temporal edges $E$, where $E \subseteq V \times V \times \mathbb{R}$. $t$ is the time of the temporal edge $(u, v, t)$.*

Our approach centers around our proposed notion of 'time-decayed line graphs' (TDLGs) which are derived from temporal graphs. Our notion of TDLGs extends the well-established idea of line graphs, which are derived from static (i.e., non-temporal), undirected graphs; the line graph of an undirected, unweighted graph is another undirected, unweighted graph that represents adjacencies between edges in the original graph. We propose to incorporate time information by using it to set the weights of the resulting line graph. Specifically, given a temporal graph $G$, we construct a TDLG, which is a static weighted line graph $L_{\text{TD}}(G)$, as follows in Definition 2.

**Definition 2** (Time-Decayed Line Graph). *Given a temporal graph $G = (V, E)$, the associated time-decayed line graph is $L_{TD}(G) = (V_L, E_L, w)$, where $V_L = E$, $E_L = \{((u, v, t_1), (v, z, t_2)) : (u, v, t_1), (v, z, t_2) \in E\}$, and the edge weight function $w : E_L \rightarrow \mathbb{R}_+$ evaluated on edge $((u, v, t_1), (v, z, t_2))$ is given by $\exp\left(-\frac{1}{2\sigma_t^2}(t_1 - t_2)^2\right)$ for some fixed $\sigma_t > 0$.*

Thus, proximity of two temporal edges in the TDLG incorporates both topological proximity in the original graph as well as proximity in time. The parameter $\sigma_t$ controls how quickly proximity in the TDLG decays as difference in time grows. For a graph with $n$ nodes and $m$ edges, we can construct the weighted adjacency matrix $A \in \mathbb{R}_+^{m \times m}$ of the TDLG as follows. Given the incidence matrix

$B \in \{0,1\}^{n \times m}$, each column vector of which is $n$-dimensional, corresponds to an edge, and is 1 for the edge's two endpoint nodes and 0 elsewhere; and also given the vector of times of the temporal edges $t \in \mathbb{R}^m$:

$$A_{ij} = \left( b_i^\top b_j \right) \cdot \exp \left( -\frac{(t_i - t_j)^2}{2\sigma_t^2} \right), \tag{1}$$

where $b_i$ and $b_j$ denote columns $i$ and $j$ of $B$. Note that $A$ is symmetric. While self-loops in line graphs are generally removed, we keep them here for simplicity. The choice of Gaussian weight decay is somewhat arbitrary, and, in theory, the TDLG concept would also work with, e.g., Laplacian weight decay. We use Gaussian decay since it is well known and effective in practice.

We now describe our approach for temporal edge embedding, classification, and link prediction.

**Temporal edge embedding and classification** For downstream tasks, we seek temporal edge embeddings, that is, an informative real-valued vector for each temporal edge that can be provided to a classifier. For temporal edge $i$, we simply return the $i^{\text{th}}$ row of the TDLG adjacency matrix $A$. This returns an $m$-dimensional sparse vector as the feature vector. Note that we could reduce the dimensionality of these vectors, e.g., via eigendecomposition, but we find that this is not necessary. We use the standard supervised learning pipeline for edge classification. Given training data comprising a set of temporal edges and the class labels of some fraction of these edges, the TDLG matrix $A$ is created using all of the edges, then a logistic regression classifier is trained using the edges for which the classes are known and the corresponding rows of $A$ as feature vectors. After this, the classifier can be used on the remaining rows to make predictions for the remaining edges.

**Temporal link prediction** We describe the full experimental setup for temporal link prediction in Section 6, but essentially, it amounts to binary temporal edge classification where the classes are $+1$ for true/positive edges $(u_i, v_i, t_i)$ and $-1$ for false/negative edges $(u_i', v_i', t_i')$. The difference is that for temporal link prediction, we must additionally be able to form representations for negative edges, as well as for test edges (which are not given when training the classifier). Let subscripts $r$ and $e$ denote training and test edges, respectively. We deal with negative edges exactly the same as positive edges besides the class labels: the positive and negative training edges are concatenated, producing the training incidence matrix $B_r \in \{0,1\}^{n \times m_r}$ and times vector $t_r \in \mathbb{R}^{m_r}$, from which we construct the training TDLG adjacency matrix $A_{rr} \in \mathbb{R}^{m_r \times m_r}$ according to Equation 1. The classifier can then be trained using the rows of $A_{rr}$ as feature vectors and the edge labels (i.e., positive/negative edge). Now feature vectors for the test edges must each be $m_r$-dimensional and consider only proximity to training edges. Letting the test incidence matrix and test times vector be $B_e \in \{0,1\}^{n \times m_e}$ and $t_e \in \mathbb{R}^{m_e}$, the test edge feature vectors will be the rows of the matrix $A_{er} \in \mathbb{R}^{m_e \times m_r}$, the $(i,j)$-th entry of which is given by

$$(b_{e\,i} \cdot b_{r\,j}) \cdot \exp \left( -\frac{(t_{e\,i} - t_{r\,j})^2}{2\sigma_t^2} \right),$$

which can be passed to the classifier for inference.

## 3 DEMONSTRATIVE EXAMPLE

To demonstrate the power of our TDLG method at capturing latent structure in temporal graphs, we introduce a simple, natural model for temporal networks based on the stochastic block model (SBM) (Holland et al., 1983). Our model is called the union of Gaussian-timed SBMs (UGT-SBM). As the name suggests, it comprises the union of several SBMs, where a normal distribution is attached to each SBM, and the edges drawn from an SBM are given a time which is sampled from its distribution. The formal defintion follows in Definition 3.

**Definition 3** (UGT-SBM). *Consider a set of $n$ nodes partitioned into two equal-sized communities $\{U, V\}$. For some integer $\Delta > 0$ and real numbers $0 < \alpha_1, \alpha_2 < 1$, construct a random temporal edge set over this graph as follows: Let there be $\frac{\Delta \cdot n}{2}$ temporal edges in each of two time periods; the times associated with edges in each time period are drawn from the normal distributions $\mathcal{N}(\mu_1, \sigma_1^2)$ and $\mathcal{N}(\mu_2, \sigma_2^2)$, respectively. Of the edges in time period 1, let $(1 - \alpha_1) \cdot \frac{\Delta \cdot n}{2}$ edges be drawn uniformly at random from $U \times V$, and let the remaining $\alpha_1 \cdot \frac{\Delta \cdot n}{2}$ edges be drawn half each from $U \times U$ and $V \times V$. Similarly, of the edges in time period 2, let $(1 - \alpha_2) \cdot \frac{\Delta \cdot n}{2}$ edges be drawn from $U \times V$, and the remaining $\alpha_2 \cdot \frac{\Delta \cdot n}{2}$ edges be drawn half each from $U \times U$ and $V \times V$.*

Here $\Delta$ is the expected degree of each node in each of the two SBMs (so the total degree of each node is $2 \cdot \Delta$), and $\alpha_1, \alpha_2$ represent the fraction of edges in each SBM which are intra-community as opposed to inter-community. This definition can generalize straightforwardly to a union of an arbitrary number of SBMs, potentially with unequal community sizes. We note some related prior models with more sophisticated temporal dependencies based on point processes: the Hawkes IRM (Blundell et al., 2012), the block point process (BPP) model (Junuthula et al., 2019), and the community Hawkes independent pairs (CHIP) model (Arastuie et al., 2020); the latter two in particular are also based on SBMs. Also related is the model from Barbillon et al. (2017), which extends SBMs not to the temporal setting, but rather to the multiplex setting (i.e., multiple graphs over the same set of nodes).

We first construct a UGT-SBM where it is impossible to distinguish any community structure if one ignores temporal information. This UGT-SBM is visualized in Figure 1. The graph has $n = 100$ nodes, with expected degrees $\Delta = 40$, and the time distributions of the two SBMs are $\mathcal{N}\left(-1, (1/2)^2\right)$ and $\mathcal{N}\left(+1, (1/2)^2\right)$; the distributions have the same variance, but one occurs earlier on average. We set $\alpha_1 = 9/10$ and $\alpha_2 = 1/10$, so that the mean fraction of intra-community edges is $\frac{\alpha_1 + \alpha_2}{2} = 1/2$; this ensures that it is impossible to retrieve community structure without considering time, as is best illustrated in Figure 1 in the union of the two SBMs, which is simply an Erdős–Rényi graph.

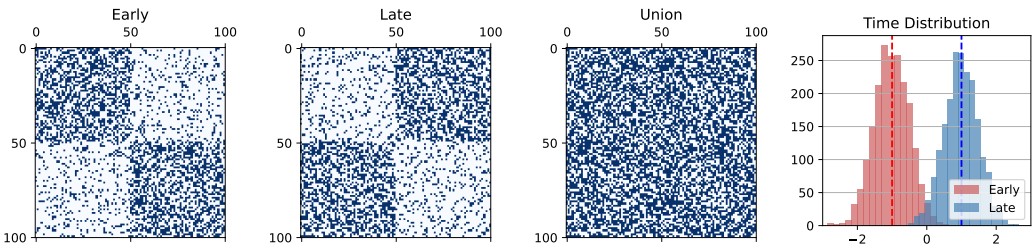

Figure 1: The motivating synthetic temporal network. The adjacency matrix of the early and late edges, as well as their union, then the distribution of times separated by the early and late edges. Note that the early edges are mostly in-group, whereas the late edges are mostly out-group.

We will show that our TDLG method can utilize the time information to recover the node communities. Since our method produces embeddings for temporal edges, to demonstrate recovery of node communities, we must aggregate the edge embeddings into node embeddings. Node embeddings can be straightforwardly constructed from edge embeddings as follows: for each node, return the mean of the embeddings of all edges incident on that node. This is described formally in Definition 4:

**Definition 4** (Mean-Edge Node Embeddings). *Suppose a network has $n$ nodes and $m$ edges, with incidence matrix $\boldsymbol{B} \in \mathbb{R}^{n \times m}$. Let $\tilde{\boldsymbol{B}}$ be the matrix that results from dividing each row of $\boldsymbol{B}$ by its sum. Given a matrix of $k$-dimensional edge embeddings $\boldsymbol{Y} \in \mathbb{R}^{m \times k}$, produce a matrix of node embeddings $\boldsymbol{X} \in \mathbb{R}^{n \times k}$ via the matrix product $\boldsymbol{X} = \tilde{\boldsymbol{B}}\boldsymbol{Y}$.*

In Figure 2, we show the result of applying our method to this UGT-SBM. In particular, we construct the TDLG with $\sigma_t = 1/2$ and show the TDLG adjacency matrix, temporal edge embeddings generated from this matrix, and node embeddings generated by Definition 4. Note that while we generally use the raw rows of the TDLG adjacency matrix as edge embeddings (which produces sparse vectors), only in this section, for visualization purposes, we instead use dense eigenvector embeddings. The embedding visualizations show that our method is recovering the community structure: The edge embeddings roughly separate the 6 kinds of edges (i.e., edges in $U \times U$, $V \times V$, and $U \times V$ occurring in each of the two time periods), and further, the node embeddings linearly separate the two node communities $U$ and $V$.

## 4 THEORETICAL DISCUSSION

We now discuss our theoretical results, which concern the action of our TDLG method on the proposed UGT-SBM family of synthetic graphs. We present these results not only to show the power of the TDLG approach, but also to contrast with deep methods, the complexity of which often precludes explicit analysis. In particular, under certain strong assumptions, our approach's simplicity allows us to give the exact expectation for the TDLG adjacency matrix $\boldsymbol{A}$ for UGT-SBMs. Further,

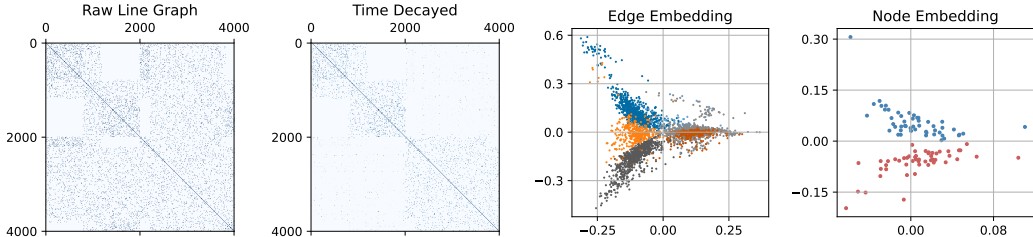

Figure 2: Applying our method to the motivating synthetic network of Figure 1. The line graph is ordered so that the early edges appear first. The embedding plots show the top second and third eigenvectors (by eigenvalue magnitude) of the TDLG adjacency. Note that edge types are roughly separated in the edge embeddings, and node types are linearly separated in the node embeddings.

we can show constructively that the TDLG method is able to preserve and disentangle the node community structure of the UGT-SBM: specifically, taking the temporal edge embeddings returned by our method and aggregating them into node embeddings via averaging over a node's incident edges (as in Definition 4) yields node embeddings which can distinguish the two node communities. The assumptions for these two results are as follows: we analyze using the expected TDLG adjacency matrix and the expected incidence matrix, and we assume zero time variance within time periods of the UGT-SBM; these assumptions all simplify the analysis by reducing variance, though perhaps via concentration bounds one could extend such results to the sampled setting (Spielman, 2012). Also, we will consider UGT-SBMs in the limit as $n \to \infty$ for simplicity of the resulting constant values, but the following analysis would carry through in general with different values.

**Proposition 4.1** (Structure of TDLG Adjacency Matrix for UGT-SBMs). *Suppose $\boldsymbol{A} \in \mathbb{R}^{m \times m}$ is the expected TDLG adjacency matrix of a UGT-SBM graph with node communities $\{U, V\}$, time periods $\{1, 2\}$, and zero time variance ($\sigma_1^2 = \sigma_2^2 = 0$). $\boldsymbol{A}$ has a $6 \times 6$ block structure where each block is constant-valued, that is, a multiple of an all-ones matrix $\boldsymbol{J}$ of some dimensionality. Letting subscripts denote time period, the block matrix is indexed by 6 edge sets: $(U \times U)_1$, $(V \times V)_1$, $(U \times V)_1$, $(U \times U)_2$, $(V \times V)_2$, and $(U \times V)_2$. In the limit as $n \to \infty$, the constant values of each block are given by the Kronecker product*

$$
\begin{array}{c}
U \times U \\
V \times V \\
U \times V
\end{array}
\left(
\begin{array}{c|c|c}
8/n & 0 & 4/n \\
\hline
0 & 8/n & 4/n \\
\hline
4/n & 4/n & 4/n
\end{array}
\right)
\quad \otimes \quad
\begin{array}{c}
1 \\
2
\end{array}
\left(
\begin{array}{c|c}
1 & \gamma \\
\hline
\gamma & 1
\end{array}
\right),
$$

*where $\gamma = \exp\left(-\frac{(\mu_1 - \mu_2)^2}{2\sigma_t^2}\right)$.*

Note that writing $\boldsymbol{A}$ as a Kronecker product involves some abuse of notation, and the actual sizes of the edge sets (and hence the sizes of the blocks) are as given in Definition 3. In the order of Proposition 4.1, these sizes are $\alpha_1 \cdot \frac{\Delta \cdot n}{4}$, $\alpha_1 \cdot \frac{\Delta \cdot n}{4}$, $(1 - \alpha_1) \cdot \frac{\Delta \cdot n}{2}$, $\alpha_2 \cdot \frac{\Delta \cdot n}{4}$, $\alpha_2 \cdot \frac{\Delta \cdot n}{4}$, and $(1 - \alpha_2) \cdot \frac{\Delta \cdot n}{2}$.

*Proof.* We calculate an entry $\boldsymbol{A}_{ij}$ of this matrix. We first consider the effect of the time decay, then the effect of edge adjacency (i.e., $\boldsymbol{b}_i^\top \boldsymbol{b}_j$). Given the assumption of zero time variance, there are only two possible values for each temporal edge's time, $\mu_1$ and $\mu_2$. This means effect of time decay on each entry of $\boldsymbol{A}$ is multiplication by either 1, if $i$ and $j$ occur at the same time, or otherwise $\gamma$.

Second, we calculate the expected value of $\boldsymbol{b}_i^\top \boldsymbol{b}_j$, that is, the expected number of endpoint nodes shared between edges $i$ and $j$. Suppose both $i$ and $j$ are in $U \times U$. Since there are $n/2$ nodes in $U$, the probability of any two random nodes in $U$ being the same is $2/n$. There are 4 pairs of endpoints in $U$ between $i$ and $j$; in the assumed $n \to \infty$ limit, the 4 events of these pairs each having the same two nodes tend toward independence, so $\boldsymbol{b}_i \cdot \boldsymbol{b}_j = 4 \cdot 2/n = 8/n$. By similar logic, if $i$ is in $U \times U$ and $j$ is in $U \times V$, there are 2 pairs of endpoints that could be identical (each of $i$'s nodes in $U$ with the single node of $j$ in $U$), so $\boldsymbol{b}_i \cdot \boldsymbol{b}_j = 2 \cdot 2/n = 4/n$. The same holds if both $i$ and $j$ are in $U \times V$. Finally, if $i$ is in $U \times U$ and $j$ is in $V \times V$, there is zero chance of a shared endpoint. Remaining combinations follow by symmetry. Combining the terms for time decay and edge adjacency, the expected TDLG adjacency matrix has the specified $6 \times 6$ block structure. ∎

We now state the second result, about the ability to distinguish UGT-SBM node communities using the TDLG method. This result requires that it is not the case that $\alpha_1 = \alpha_2 = 1/2$; in this case of

UGT-SBM, the SBMs at both time periods are Erdős–Rényi, so there is no community structure to recover.

**Proposition 4.2** (TDLG Embedding Recovers Communities from UGT-SBMs). *Suppose $\boldsymbol{A} \in \mathbb{R}^{m \times m}$ is the expected TDLG adjacency matrix of a UGT-SBM graph with node communities $\{U, V\}$, time periods $\{1, 2\}$, and zero time variance ($\sigma_1^2 = \sigma_2^2 = 0$). Let $\boldsymbol{X} \in \mathbb{R}^{n \times m}$ be the mean-edge node embeddings resulting from treating the rows of $\boldsymbol{A}$ as edge embeddings. Assuming that $\gamma \neq 1$ and it is not the case that $\alpha_1 = \alpha_2 = 1/2$, the node communities are distinguishable in $\boldsymbol{X}$.*

For brevity, the proof is deferred to Appendix A.1. It essentially involves the same flavor of counting-based arguments as Proposition 4.1, though it is much more involved. As part of this proof, we provide a description of part of the expected block structure of the resulting node embeddings $\boldsymbol{X}$, similar to the preceding description of the TDLG adjacency matrix $\boldsymbol{A}$. When time decay is not applied (i.e., by setting $\gamma = 1$), we find that nodes in $U$ and $V$ become indistinguishable in the derived embeddings for UGT-SBMs with $\alpha_1 + \alpha_2 = 1$, as opposed to just when $\alpha_1 = \alpha_2 = 1/2$. These are UGT-SBMs like the one from Figure 1, which do have community structure when considering time, but become Erdős–Rényi graphs when ignoring time.

## 5 RELATED WORK

**Node embeddings** Many methods have been proposed over the years for the node embedding task. The best understood are the classical 'spectral' methods based on eigendecomposition of the graph's adjacency matrix or Laplacian (Roweis & Saul, 2000; Belkin & Niyogi, 2001). Perhaps the most famous non-spectral method is DeepWalk (Perozzi et al., 2014), which takes random walks on the graph, then, using a nonlinear, probabilistic objective, increases the similarity of the embeddings of nodes which frequently co-occur in the walks. Similar methods include node2vec (Grover & Leskovec, 2016), which adds bias to the random walks, and LINE (Tang et al., 2015), which increases scalability. GraRep (Cao et al., 2015) and NetMF (Qiu et al., 2018) incorporate both matrix factorization like the classical methods and nonlinear objectives like the recent ones, yielding good speed and performance. Unsupervised deep learning approaches, like VGAE (Kipf & Welling, 2016) and SDNE (Wang et al., 2016), involve deep autoencoders; besides this, there is a wide array of supervised deep models which implicitly form node representations, including graph convolutional networks (GCNs) (Kipf & Welling, 2017) and graph attention networks (GATs) (Veličković et al., 2018).

**Edge embeddings** The task of creating vector representations for each edge, rather than each node, has seen less exploration. A common strategy is to simply create node embeddings, then process them into edge embeddings by aggregating the embeddings of the two endpoints of an edge. In particular, this is done by taking the mean, entrywise product, cross product, or concatenation of the two node embedding vectors. Notably, node2vec uses this strategy for link prediction; Shi et al. (2018) and Verma et al. (2019) are more examples where this is part of a larger framework. More along the lines of our method, though still not involving temporal networks, Bandyopadhyay et al. (2019) produce edge representations in an unsupervised manner by embedding the nodes of the line graph of the original network. The method of Li et al. (2017b) implicitly takes a similar approach by taking a random walk on the edges and iteratively updating edge embeddings, as DeepWalk does for node embeddings. Some graph deep learning approaches (Monti et al., 2018; Gong & Cheng, 2019) include layers which form edge embeddings, but do so as part of a supervised framework; by contrast, Zhou et al. (2018b) propose an unsupervised deep method based on generative adversarial networks, though their approach is fairly complicated. Also of note, though not directly applicable to the standard or temporal edge setting, several works in the area of knowledge graphs form embeddings of relations between entities (Bordes et al., 2013; Yang et al., 2014; Chen & Quirk, 2019).

**Embeddings for temporal networks** Many methods for temporal network embedding can be seen as variants of those for static graphs. Dunlavy et al. (2011) propose a variant of factorization-based approaches, intended for a dynamic network comprising snapshots of static graphs at consecutive intervals of time. It directly incorporates time information by stacking adjacency matrices of different time steps and employing tensor factorization. DANE (Li et al., 2017a) is another factorization approach, though it only factorizes matrices. To boost efficiency, rather than computing node embeddings from scratch for each snapshot, DANE iteratively updates the embeddings from the previous time step according to the change in the network; TIMERS (Zhang et al., 2018) is a similar approach which analyzes the growth of the error over iterative updates to determine when it is

Table 2: Statistics of datasets used in our experiments.

| Dataset | # Nodes | # Edges | Time Span (days) |
|---|---|---|---|
| BITCOINALPHA | 3783 | 24186 | 1901 |
| BITCOINOTC | 5881 | 35592 | 1903 |
| ESCORTS | 10106 | 50632 | 2232 |
| WIKIELECT | 7118 | 107071 | 1378 |
| EPINIONS | 131828 | 841372 | 943 |

necessary to re-embed from scratch. NetWalk (Yu et al., 2018) is a variant of random walk-based methods like DeepWalk; along the same lines, it saves time by efficiently updating embeddings at each snapshot using warm starts and reservoir sampling. CTDNE (Nguyen et al., 2018) is another random walk method which incorporates time information using the constraint that the walks must obey time. Notably, like our method, CTDNE is designed for networks with continuous time rather than just snapshots of different time intervals.

Among unsupervised deep approaches, DynGEM (Goyal et al., 2018) is an autoencoder method which again uses warm starts for each snapshot, both for speed and to encourage embeddings to be approximately preserved across time steps; DynGraph2Vec (Goyal et al., 2020) more directly allows information at different snapshots to be integrated by employing an RNN to evolve representations across time steps. The similarly-named DyLink2Vec (Rahman et al., 2018) is another deep autoencoder approach, which also integrates information across time steps; notably, like our method, it yields embeddings for links rather than nodes, though it works on snapshots rather than continuous time. More recently, EvolveGCN (Pareja et al., 2020) proposes the interesting idea of using an RNN to evolve not the node embeddings, but the weights of a GCN model which generates embeddings for each snapshot. Some deep methods also handle continuous time: using point processes, Know-Evolve (Trivedi et al., 2017), HTNE (Zuo et al., 2018), and DyRep (Trivedi et al., 2019) model the occurrence of temporal edges, while Dynamic-Triad (Zhou et al., 2018a) aims to capture structural information by modeling the closure of wedges into triangles over time. More recent methods handling continuous-time include TGAT (Xu et al., 2020), TGNs (Rossi et al., 2020), MeTa (Wang et al., 2021), and PINT (Souza et al., 2022); these methods adapt and integrate various recent deep learning modules and concepts into the temporal network setting, such as self-attention, memory, data augmentation, and positional embedding. The PINT paper in particular performs an analysis of the continuous-time and discrete-time settings, proving that the latter setting reduces to the former without loss of information, whereas the converse is not always true, and hence the class of methods handling continuous-time directly is theoretically more powerful in this sense.

Of the many approaches for temporal networks that we have discussed, most are principally node embedding methods from which edge embeddings can be generated by aggregation; to our knowledge, ours is the first work which directly forms embeddings for continuous-time temporal edges without supervision and without aggregating node embeddings.

# 6 EXPERIMENTS

We evaluate our method on two kinds of tasks, edge classification and temporal link prediction, on a collection of five real-world temporal network datasets. The statistics for these networks are provided in Table 2, but we defer discussion of these datasets to Appendix A.2.

## 6.1 EDGE CLASSIFICATION

**Experimental setup** We use a logistic regression classifier to which the input is a feature vector for each edge of the network and the target is a binary edge class. We make random 70%/30% splits of training/test data, and report test AUC of binary classification across 10 trials with 10 random splits.

In addition to results for our TDLG method's sparse embeddings, we report results for several other methods. We compare to results for some prior methods applied to these datasets: 1) CTDNE (Nguyen et al., 2018); 2) TIMERS (Zhang et al., 2018); 3) EvolveGCN (Pareja et al., 2020); 4) DynGEM (Goyal et al., 2018); 5) GCRN (Seo et al., 2018); and 6) VGRNN (Hajiramezanali et al., 2019). Note that these methods output $128$-dimensional dense embeddings, unlike our TDLG method, which outputs $m$-dimensional sparse embeddings; for an additional, direct comparison, we also report results

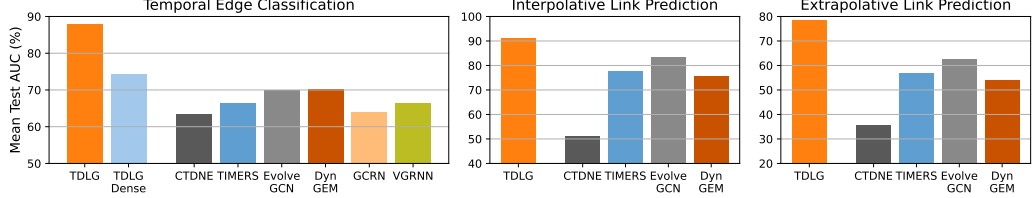

Figure 3: Test AUC on the edge classification task for real-world datasets using our TDLG method, CTDNE, TIMERS, EvolveGCN, and DynGEM.

for using the top 128 eigenvectors (i.e., those with highest magnitude eigenvalues) of the TDLG adjacency matrix ('TDLG Dense'). With the exception of the methods we introduce and CTDNE, all other methods do not directly handle the continuous time stamps of the datasets; for these methods, we slot the edges into discrete snapshots by evenly dividing the full time span into 20 intervals. All experiments are run on an Xeon Gold 6130 CPU and Tesla v100 16GB GPU; only the latter four methods (i.e., the deep methods) use the GPU, and we are unable to run these four methods on EPINIONS due to GPU memory limitations. We release code in the form of a Jupyter notebook (Pérez & Granger, 2007) demo which is available at github.com/schariya/tdlg.

**Hyperparameter selection** For all embedding methods, we use the scikit-learn (Pedregosa et al., 2011) implementation of logistic regression; we increase the maximum iterations to $10^3$, and, since the edge classes are generally imbalanced across the datasets, we set the class_weight option to 'balanced,' which adjusts loss weights inversely in proportion to class frequency. We keep otherwise default options. For our TDLG method, we set the time scale hyperparameter $\sigma_t$ as ratio of the standard deviation of the edges' times; calling this standard deviation be $\sigma_T$, we use $\sigma_t = 10^{-1} \cdot \sigma_T$, which is chosen by informal tuning. In Appendix A.3, we explore the impact of varying $\sigma_t$. We use the implementation of Liu et al. (2020) for TIMERS and all deep methods, all with default hyperparameters. For the EPINIONS dataset only, for our TDLG method, we modify the solver for sparse logistic regression to the 'saga' solver, which is more scalable than the default 'lbfgs' solver, and reduce the maximum number of iterations to 100 (the default value).

**Results** We first plot mean AUC and 95% confidence intervals for selected methods - our TDLG, CTDNE, TIMERS, EvolveGCN, and DynGEM - on all datasets in Figure 3. Our TDLG method achieves higher performance than the comparison methods on all datasets, often by a large margin.

Figure 4: Test AUC on all three tasks, aggregated across datasets: mean performance over the evaluated real-world datasets, excluding EPINIONS.

To compactly compare all 9 of the methods across these datasets, we aggregate the performance across the datasets as follows: we plot the mean across all datasets, excluding EPINIONS, of each method's test AUC. We exclude EPINIONS since we are unable to run some methods on this dataset. See Figure 4 (left). TDLG achieves the highest performance out of all these methods on all datasets; the dense variant of TDLG, which perhaps provides more direct comparison with other methods, sees reduced performance, but also outperforms all prior methods. Tables of full experimental results are deferred to Appendix A.5.

**Runtime** We evaluate the time efficiency of our approach against the selected baseline methods. For this, we report the mean runtime in seconds for one trial of edge classification. Runtime constitutes one trial of learning embeddings, learning a logistic regression classifier on training data, and predicting on test data. See Figure 5. Across all five datasets, our TDLG method is the fastest, often by a significant margin. We discuss the scalability of our method in Appendix A.4.

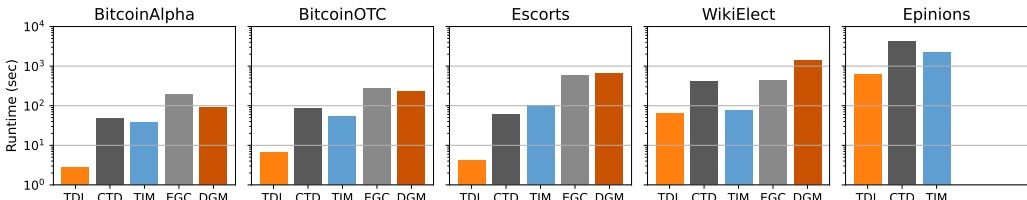

Figure 5: Runtime (seconds) on the edge classification task.

## 6.2 LINK PREDICTION

**Experimental setup**  Given the dataset of actual, 'positive' edge tuples $(u_i, v_i, t_i)$, we generate an equal-sized set of fake, 'negative' edges by independently shuffling the three columns (i.e., independently shuffling the set of first nodes, the set of second nodes, and the times). Note that this preserves the distribution nodes/times in of each of the columns. The task is to distinguish positive edges from negative ones. We evaluate two different settings for link prediction. In the first, which we call temporal link outlier detection, the test edges are in the same time interval as the training edges. In the second, which we call temporal link prediction, the test edges are in an interval of time following that of the training data. We also call these settings 'interpolative' and 'extrapolative,' respectively. Prior work generally focuses on the latter setting, since it is more applicable, e.g., for predicting future events/behavior, but we also evaluate the former mainly out of scientific interest.

Specifically, given that we are splitting the data into 20 intervals, we consider intervals 1-19 to be 'early' data and interval 20 to be 'late' data. To share computational effort across the two settings, we train both classifiers on 70% of edges (both positive and negative) from the early intervals. For the interpolative setting, the test data comprises the remaining 30% of edges from the early intervals. For the extrapolative setting, the test data comprises all edges from the late interval. Evaluating the link prediction task has greater computational cost compared to edge classification since the input graph changes across trials rather than just train/test splits of labels, requiring new embeddings to be made for each trial. For this reason, we only test the 5 selected methods from the previous section. We report mean AUC and 95% confidence intervals across 5 trials, each trial having different random shuffles producing different negative edges, as well as different 70%/30% train/test splits.

**Hyperparameter selection**  We generally use the same hyperparameters as for edge classification, with one exception: since there is an equal number of positive and negative edges, we no longer set the class_weight option to 'balanced,' and all edges are given equal weight.

**Results**  As for edge classification, we plot the aggregated (mean) performance across the datasets in Figure 4 (right), for both the interpolative and extrapolative settings of link prediction. For brevity, we defer non-aggregated and tabular results to Figure 8 in Appendix A.5. In both settings, across all datasets, our TDLG method achieves the highest performance, often by a large margin. We do find that the deep methods EvolveGCN and DynGEM generally perform better than CTDNE and TIMERS, and EvolveGCN in particular is competitive with TDLG on some datasets.

## 7 CONCLUSION

We present a novel framework for temporal edge embedding called the time-decayed line graph (TDLG). Unlike some prior methods, our method works directly with continuous timestamps rather than requiring discretization of times, and directly generates temporal edge embeddings rather than requiring aggregation of node embeddings. Also in contrast to many prior methods, ours has a simple concept and high ease of implementation, since it essentially just creates a sparse matrix of proximities between temporal edges. Despite its simplicity, our method achieves superior performance for several downstream tasks on benchmark temporal network datasets, while requiring less runtime. Its simplicity also facilitates theoretical guarantees of its effectiveness on our proposed UGT-SBM family of networks. Future directions include extensions of our approach to more general scenarios involving, for example, streaming input data or inductive learning; an approach for the latter could be as straightforward as running an inductive method on the TDLG of the input graph as opposed to the raw graph itself. Broadly, this work highlights the potential of methods involving engineered features, as opposed to learned ones, to achieve solid performance by directly invoking simple inductive priors.

## ACKNOWLEDGMENTS

Cameron Musco and Sudhanshu Chanpuriya were partially supported by an Adobe Research grant.

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
