# OpenReview forum: "Direct Embedding of Temporal Network Edges via Time-Decayed Line Graphs"
_ICLR.cc/2023/Conference — ICLR 2023 poster_

### Official Review · Reviewer_qEEe · 2022-10-30

**Confidence:** 5
**Correctness:** 3
**Technical Novelty And Significance:** 4
**Empirical Novelty And Significance:** 3
**Recommendation:** 8

**Clarity, Quality, Novelty And Reproducibility:**

- Clarity: Very High. An absolute joy to read! I have only a few suggestions to improve the presentation above.
- Quality: High. I noticed only some minor issues with notation. The experiments are sound. The theoretical analysis is a bit simplistic.
- Novelty: Very High. This is the first approach I have seen for edge embeddings in temporal networks, and it involves the creation of a time-decayed line graph, which is also a novel concept.
- Reproducibility: Very High. Between the attached Jupyter notebook in the supplementary material and the descriptions in the paper, I am confident that I could reproduce the results. The simplicity of the proposed approach helps further here--there is no dependency on tons of additional libraries.


**Strength And Weaknesses:**

Strengths:
- Highly creative and simple approach. While line graphs have previously been used in various network science applications, the proposed time-decayed line graph (TDLG) representation appears to be novel. The edge embedding is simply a row of the TDLG adjacency matrix.
- The authors prove that their proposed embedding can separate nodes in different communities in a simplified temporal stochastic block model (TSBM).
- Strong performance in experiments compared to recent temporal network embedding methods, with better accuracy while maintaining low runtime due to the simplicity.
- Top-notch presentation--very well written and a joy to read. Thank you!

Weaknesses:
- Theoretical analysis applies only to the expected adjacency matrix and also assumes no variance in the edge timestamps. While this is a good start, it is too simplistic to be of practical use. As the authors suggest, the results could likely be extended to the sampled setting using concentration inequalities for matrices. This type of analysis was conducted on the Community Hawkes Independent Pairs (CHIP) model--another type of temporal SBM model involving Hawkes processes. See Arastuie et al. (NeurIPS, 2020) for more details.
- Naming the demonstrative example in Section 3 a "temporal SBM" is somewhat misleading. There are lots of different probabilistic generative models for temporal networks in the literature that can be considered temporal SBMs, including the CHIP model and even older work such as the Hawkes IRM (Blundell et al., NeurIPS 2012). I suggest the authors to pick a more specific name for their proposed simple model and add references to other work on temporal SBMs.

References:
- Arastuie, M., Paul, S., & Xu, K. S. (2020). CHIP: A Hawkes process model for continuous-time networks with scalable and consistent estimation. In Advances in Neural Information Processing Systems 33 (pp. 16983-16996).
- Blundell, C., Heller, K. A., & Beck, J. (2012). Modelling reciprocating relationships with Hawkes processes. In Advances in Neural Information Processing Systems 25 (pp. 2600-2608).

Questions:
1. In the specification of the temporal SBM, it appears that the number of temporal edges is fixed to $\Delta \cdot n/2$. This does not match with the usual definition of the SBM, where the number of edges is not fixed but the probability of forming an edge between any two communities are fixed. This is analogous to the difference between a $G(n,m)$ and $G(n,p)$ random graph model. How are you fixing the number of edges while maintaining community structure?
2. One disadvantage of using line graphs is the large size of the adjacency matrix they generate, which requires lots of memory to store even using sparse matrices. It appears that your TDLG also has this disadvantage. How much main memory is required to conduct your experiments, especially on the large Epinions data? Do you further sparsify your TDLG in any way, e.g., by removing edges with weights below a certain threshold?

Typos and minor issues:
- A figure showing the time-decayed line graph for a toy example would be helpful to illustrate the construction of a time-decayed line graph. I was already familiar with line graphs before reading this paper, so I had no problem understanding the definition, but I suspect many readers may not have previously seen line graphs. To create some room, I suggest moving the proof of Proposition 4.1 also to the appendix.
- The use of $\sigma_1, \sigma_2$ in the temporal SBM is potentially confusing given the use of $\sigma_t$ for the Gaussian decay. I would suggest changing $\sigma_t$ to a different symbol.
- I think there is some notation inconsistency in Proposition 4.1. $\gamma$ is defined but never used, and $\lambda$ is used in the Kronecker product but not defined.


**Summary Of The Paper:**

The authors propose an approach for computing edge embeddings in temporal networks evolving in continuous time. By considering direct embeddings of edges rather than nodes, the authors claim that they can achieve better accuracy on downstream tasks such as edge classification. The proposed edge embedding uses a time-decayed line graph (TDLG) constructed from the original temporal network. The authors demonstrate impressive gains over other temporal network embedding methods for edge classification and link prediction on several real networks. They also present some theoretical analysis on their proposed embedding approach.

**Summary Of The Review:**

The authors have proposed a highly novel and simple approach for temporal network embedding. The simplicity enables theoretical analysis of the model. It also has very strong empirical performance. I could see this paper having tremendous impact on the ever-growing community interested in graph embeddings. It could also inspire more useful rigorous analysis of the proposed model. I strongly support this paper for acceptance!

---

> ### Author Response · Authors · 2022-11-18
> **Response to Reviewer qEEe**
>
> Thank you for the very positive review.
>
> ### Question 1, on the Temporal SBM’s fixing the number of edges
>
> Indeed, our definition of the TSBM is analogous to the $G(n,m)$ model rather than the $G(n,p)$ model, which is how SBMs are usually defined. The two models can be quite similar in certain limits (i.e., as node degrees concentrate around the expectation), and we find that the results for the experiment on the synthetic graph in Section 3 are essentially the same with either one. Using the $G(n,m)$ version facilitates the theoretical analysis, since fixing the number of edges also fixes the size of the line graph adjacency matrix.
>
> > How are you fixing the number of edges while maintaining community structure?
>
> Definition 3 for the TSBM essentially gives a constructive algorithm. Ignoring time for now and calling the node partition (the two communities) $U,V$, we fix some total number of edges, then determine a partition of these edges which will be placed into $U \times U$, $V \times V$, and $U \times V$. For example, in the purely assortative case ($p=1,q=0$ in the usual notation), if we want $100$ edges in total, we would have $50$, $50$, and $0$ edges, respectively, in each community pair. The sampling takes place after this step; for example, for each of the $50$ edges to be placed in $U \times U$, we must sample a node pair $u_1,u_2 \in U$ to hold the edge. This example demonstrates strong community structure, since there are no edges across the cut between $U$ and $V$.
>
> ### Question 2, concerning scalability
>
> Thank you for raising this important point. We have now included a discussion on the scalability in a new subsection of the appendix, and restate a bit as follows. \
> The number of nonzero elements in a line graph scales with the sum of squared degrees of the nodes. In theory, this could be very expensive, but real-world degree distributions generally seem to make this practical, at least up to the fairly large Epinions dataset of over 800K edges. We are able to run our method on Epinions on a standard 16GB laptop, which is not true of many of the comparison methods. At your suggestion, we include a plot of the number of nonzero elements in the TDLG matrices of the real-world graphs, relative to the sizes of the embeddings of other methods, and find that they are generally comparable.
>
> > Do you further sparsify your TDLG in any way, e.g., by removing edges with weights below a certain threshold?
>
> No, we do not employ this kind of sparsification, or any other methods to lower the memory usage, since it is unnecessary for the datasets we consider. However, this is an interesting idea. We have added a note of it, along with a few other possible strategies for scaling to much larger datasets, in the new subsection on scalability.
>
> ### On prior synthetic models for temporal networks
>
> Thank you for directing us to the work on the Hawkes IRM and CHIP. Where we introduce our model in the paper, we have now integrated some discussion of these two papers along with some others, and are excited to look further into them. We have also renamed the Temporal SBM to something more specific – the Union of Gaussian-Timed SBMs (UGT-SBM).
>
> ### Minor issues
>
> Thank you for pointing out the typo in Proposition 4.1 – it has been corrected.
>
> > A figure showing the time-decayed line graph for a toy example would be helpful to illustrate the construction of a time-decayed line graph.
>
> This is a great suggestion – indeed, a diagram might be useful to those unfamiliar with line graphs, likely more so than the plots of the adjacency matrices in Figure 2. We are a bit too limited on time to implement this suggestion for now, but we will strongly consider it for a final version if accepted.

---

> > ### Comment · Reviewer_qEEe · 2022-12-02
> > **Thanks for the clarifications**
> >
> > Thank you for providing the clarifications! Following the discussions between the authors and other reviewers, I have decided to lower my score to 8 (accept), rather than 10 (strong accept). I still strongly support the paper, but I perceive the significance to the ICLR community might be a bit lower. For example, see the discussions on embeddings for transductive vs. inductive link prediction.

---

> > > ### Author Response · Authors · 2022-12-03
> > > **Response to Follow-up**
> > >
> > > Thank you for following up and explaining your rationale. To collect and briefly clarify some points on transductive vs inductive settings, in case it is of interest to you or other reviewers: As presented in this paper, our method is designed to work for the transductive setting, with training and test edges arising from a single network, rather than the inductive setting, which can involve separate networks. We also only evaluate on the transductive setting in this paper (though, to make a distinction, one task we do evaluate on is "extrapolative" link prediction, where the test temporal edges have times which are strictly after the training edges). While there is a lot of interesting recent work on the inductive setting, some of which has been mentioned by other reviewers, the transductive setting also sees much interest. See, for example, the many papers we compare to, most of which focus exclusively on the transductive learning.\
> > > We mention in the revised version that exploring an inductive version of our TDLG method would be an interesting extension, and this could be as straightforward as the following: simply construct the TDLG of the incoming temporal network at either training or inference time, and use one of the many inductive models for static graphs, e.g., GraphSAGE. Broadly, the TDLG idea we propose is a simple method for converting a temporal network to a static weighted network in a meaningful way (see prior discussion on the inductive bias of locality), so it can quite easily provide a baseline extension of prior methods for static networks, to temporal networks.

---

### Official Review · Reviewer_GdB7 · 2022-11-01

**Confidence:** 4
**Correctness:** 4
**Technical Novelty And Significance:** 3
**Empirical Novelty And Significance:** 3
**Recommendation:** 8

**Clarity, Quality, Novelty And Reproducibility:**

The problem is well-motivated and the empirical part of the paper is substantial. The results look promising. I believe the results are original, clearly stated, and described well along with the intuition behind the method.

**Strength And Weaknesses:**

### Strengths
1. The paper is easy to follow and well-written.
2. The proposed method is easy to implement and simple to understand, yet, is shown to perform better than prior methods via an extensive set of experiments.
3. Theoretical discussion is apt and provides the relevant intuition behind the method very cleanly.

### Weaknesses
1. The theoretical discussion is limited to a very bare-bones TSBM. The requirement of zero time variance in the results is not well-justified.
2. I believe there is a missing discussion on spectral methods for multi-graph community detection where multiple graphs are given, and one is supposed to figure out the community structure. From a theoretical standpoint, I don't see a difference between the temporal problem and the multi-graph problem. A brief discussion with references would be nice. In particular, the survey "Abbe, E., 2017. Community detection and stochastic block models: recent developments. JMLR, 18(1), pp.6446-6531" is a good reference.

### Questions and Feedback
1. [Def 3] Could you justify taking the time values in the two periods to be Gaussian? It seems that a primary motivation was to not have to discretize time. However, saying "two" periods but taking the values to be Gaussian still in some sense makes me think that the time values are discretized. I'm thinking that: Gaussians have nice tails, so they are essentially their mean (concentration), then in this regime how is the model different from one where time is discretized?
2. [Fig 1] The toy example of TSBM is very intuitive. However, it might be good to add why spectral methods are not applicable to this problem. In the setup of the figure, looking at, for example, the second eigenvector of the adjacency matrix of the graphs in both time periods separately might also work (will give community labels, where one can combine the two sets of labels in a clever way to obtain more confidence). Does the sparsity of the graph play a role here? I'm thinking that if the graph is very sparse, then spectral methods won't work.
3. [Proposition 4.1] Do $\lambda$ and $\gamma$ denote the same thing?
4. [**Important**] Why do we need time variance $0$ for the theoretical results, i.e., both Prop 4.1 and 4.2? Can you provide a brief sketch for how the proof will go for non-zero variance? I believe something can be done since it's Gaussian, otherwise, why take Gaussian to begin with? Shouldn't the definition simply say that you are taking two discrete times $\mu_1,\mu_2$ for the two periods instead of Gaussian?

**Summary Of The Paper:**

The authors present a simple framework to address two issues with learning on temporal networks: (1) manual discretization of time disregards precise time information, and (2) in general, representations of nodes are returned instead of edges, making it suboptimal for edge-classification tasks. They propose a method that constructs the line graph of the network and set the edge weights based on the differences in time between the edge-interactions. This new graph represents topological proximity (existence of the edges) and temporal proximity (weights of the edges). This appears to be the first method that directly constructs embeddings for continuous-time temporal edges without aggregating node embeddings. The method is simple and intuitive. For theoretical analysis, the authors introduce a data model, called the temporal stochastic block model (TSBM), a temporal variant of the popularly studied SBM. The method is also easy to implement and is shown to perform better than prior methods in experiments on five benchmark real-world temporal networks.

**Summary Of The Review:**

Solid empirical results. Slightly lacking theoretical arguments with very strong assumptions (zero time-variance). I believe this could be a good contribution.

---

> ### Author Response · Authors · 2022-11-18
> **Response to Reviewer GdB7**
>
> Thank you very much for the review. We address the questions and concerns below.
>
> > [Proposition 4.1] Do $\lambda$ and $\gamma$ denote the same thing?
>
> This is a typo – it should be $\gamma$ only. Thank you for pointing this out.
>
>
> > I believe there is a missing discussion on spectral methods for multi-graph community detection where multiple graphs are given, and one is supposed to figure out the community structure. From a theoretical standpoint, I don't see a difference between the temporal problem and the multi-graph problem.
>
> Thank you for bringing up this point. Indeed, the multi-graph (AKA multiplex/multilayer network) community detection setting is related to our proposed synthetic model, especially in the 'discrete’-time regime with zero time variance. We have added some discussion of the SBM-based multiplex network model of Barbillon et al. (2017). At the suggestion of Reviewer qEEe, we have also added some discussion of a few other related models for the temporal setting. We note some difference between the multiplex and temporal settings, related to the point below:
>
> > In the setup of the figure, looking at, for example, the second eigenvector of the adjacency matrix of the graphs in both time periods separately might also work.
>
> This is true, but the crux of the issue is that we do not know the partition between the edges sampled from the “early” and “late” SBMs. In the regime of low time variance and two discrete times, it is possible to trivially separate the SBMs, but our experiment in Section 3 is in the non-zero time variance regime, and our theoretical result does not conceptually rely on this trivial separation.
>
>
> > Could you justify taking the time values in the two periods to be Gaussian?
>
> > Why do we need time variance $0$  for the theoretical results, i.e., both Prop 4.1 and 4.2? Can you provide a brief sketch for how the proof will go for non-zero variance?
>
> While the use of Gaussian time distributions, and specifically the focus on the low-variance regime greatly simplifies the analysis, it is not entirely divorced from a continuous-time context. In general, the proposed TSBM is meant to concretize the idea of interactions between two communities of nodes, where the ratio of in-group to out-group interactions varies continuously over time – the Gaussian time distributions are our simple way of allowing for this continuity, but one might imagine other ways.
>
> The definition with Gaussian times (i.e., with non-zero variance) is used for all of Section 3, including the experiment, which shows empirically that TDLG can work for our synthetic TSBM graph with non-zero time variance. As you note, our current theoretical results do require zero time variance, i.e., discrete times.
>
> The idea/sketch for a theoretical result with non-zero variance would be to write the matrix as a low-rank block matrix entrywise multiplied by a random matrix, where the entries of the random matrix are determined by the time gaps between different edges (which themselves will be Gaussian.) Standard matrix concentration analysis should allow for the analysis of the eigenvectors and eigenvalues of such a matrix, showing that the eigenvectors correlate strongly with the community indicator vectors. This is, for example, how spectral methods for the standard stochastic block model are analyzed (Spielman 2012). In that case, the perturbation only comes from the sampling of edges, with no further perturbation due to random time decays. A key challenge in our setting is that the random entry perturbations are not fully independent – since the time gap between different pairs of edges will be correlated. The effect of this correlation would have to be bounded.
>
> ---
>
> Barbillon, Pierre, et al. "Stochastic block models for multiplex networks: an application to a multilevel network of researchers." Journal of the Royal Statistical Society: Series A (Statistics in Society) 180.1 (2017): 295-314.
>
> Spielman, Daniel. "Spectral graph theory." Combinatorial scientific computing 18 (2012).

---

### Official Review · Reviewer_wNXK · 2022-11-01

**Confidence:** 4
**Correctness:** 3
**Technical Novelty And Significance:** 3
**Empirical Novelty And Significance:** 2
**Recommendation:** 5

**Clarity, Quality, Novelty And Reproducibility:**

It is well written paper and easy to follow. However some parts could be explained better, e.g., explain the class imbalance of datasets in the edge classification task.



**Strength And Weaknesses:**

The performance gain is impressive and the proposed model is simple and intuitive.

However, key related works are missing, e.g. TGN: TEMPORAL GRAPH NETWORKS FOR DEEP LEARNING ON DYNAMIC GRAPHS

There are also not much discussion on DTDGs and how this method performs on them.
Some of the claims might also not be correct [although said differently in different parts of the paper]. For example, there is a statement in the abstract that "First, time is assumed to be discretized, so if the time data is continuous, the user must determine the discretization and discard precise time information", while there are many methods for continues times graphs and some recent work explicitly discuss the relation CTDG and DTDG models (this is recent so of course not listing it as a missing reference but more as a reference to see https://arxiv.org/pdf/2209.15059.pdf). This is later expanded and corrected in text, but the sentence in abstract can be rephrased.

The argument that node to edge embedding is suboptimal seems weak given the proposed method needs a vector of length m, m being the number of edges. There is an attempt to make it smaller with SVD but performance drops.

The only parameter the model has is a hyper parameter tuned informally, so the although performance is good, this is not much a representation learning.

**Summary Of The Paper:**

This paper provides an interesting approach for generating link embeddings in temporal graphs, which is converting the graph into a line graph and connecting edges (now nodes), based on the temporal difference they have in the initial graph. They show that with this representation, the adjacency matrix representation is enough to achieve very strong performance in edge prediction and classification. The method is simple and seems to outperform the state of the art representation learning models by a significant margin.

**Summary Of The Review:**

Simple way for edge representation in temporal graph, with no learning, that performs significantly better than the baselines. It seems not practical given the size of representation is number of edges in the graph.

---

> ### Author Response · Authors · 2022-11-18
> **Response to Reviewer wNXK**
>
> Thank you for the helpful suggestions and for pointing out these related papers, which we have now included in the paper.
>
> > key related works are missing, e.g. TGN: TEMPORAL GRAPH NETWORKS FOR DEEP LEARNING ON DYNAMIC GRAPHS
>
> > There are also not much discussion on DTDGs… some recent work explicitly discuss the relation CTDG and DTDG models (this is recent so of course not listing it as a missing reference but more as a reference to see https://arxiv.org/pdf/2209.15059.pdf)
>
> Thank you for bringing these works to our attention. We have added some discussion of TGN and PINT (from the second reference), along with a few other recent methods, to the related work section. We have also added a note in the paper that, as the PINT paper discusses and shows theoretically, the discrete-time setting reduces to the continuous-time setting without loss of information, whereas the converse is not always true.
>
>
> > there is a statement in the abstract that "First, time is assumed to be discretized, so if the time data is continuous, the user must determine the discretization and discard precise time information" … This is later expanded and corrected in text, but the sentence in abstract can be rephrased.
>
> Apologies for the confusion. As suggested, we have now rephrased this to be more clear. To clarify, in the abstract, we do not say that prior methods generally assume discretized time; rather, we say that prior methods generally assume discretized time OR are based on node embedding rather than edge embedding, which we believe is indeed true. To be more clear and avoid misleading any reader that prior continuous-time methods do not exist, at your suggestion, we have changed this bit in the abstract to “many methods assume time to be discretized.” As you note, we expand on this later in the related work section, listing several continuous-time (but not edge-based) methods, so we hope this solution is clear overall.
>
> > The argument that node to edge embedding is suboptimal seems weak given the proposed method needs a vector of length m, m being the number of edges. There is an attempt to make it smaller with SVD but performance drops.
>
> We note that, as Figure 4 shows, while the 'dense’ SVD version of our method (which uses the same dimensionality of embedding vectors as the comparison methods) underperforms the primary 'sparse’ version, it does still outperform the comparison methods on the temporal edge classification task.

---

> > ### Comment · Reviewer_wNXK · 2022-11-27
> > **Thank you for the response**
> >
> > The main idea of using line graphs to represent temporal graphs is also suggested in previous works,  e.g. Temporal Graph Kernels for Classifying Dissemination Processes. In Proceedings of the 2020 SIAM International Conference on Data Mining. SIAM, 496–504.
> > there are also some other works that use similar dynamic line graph expansion, which seem very relevant and can be included/cited in the paper. Overall, while I appreciate the study and I think it is interesting to be shared, I am not sure if it passes the bar for ICLR, with regards to novelty (above paper for example), methodology (seems to memorize all past edges), and impact (scalability and generalizability given the representation keeps all past edges). The paper is interesting however and I would be happy if it gets accepted based on other reviewers ratings.

---

> > > ### Author Response · Authors · 2022-11-29
> > > **Response to Follow-up**
> > >
> > > Thank you for following up. We would like to emphasize that the key concept of our work involves not only the line graph, but also the decaying of edge weights based on time difference, as well as the observation that direct application of the sparse adjacency vectors as embeddings (i.e., without any dimensionality reduction) yields good performance. Indeed, there are many prior works which use line graphs for graph machine learning, and we discuss at least one of these in the related works section. We would be happy to add some more discussion of others, such as the paper you suggest, but we note that they generally do not contain the remainder of our method's concept. Concerning your comment on methodology, we think it is a key contribution of our work to show that simply capturing locality of interactions (without modeling, e.g., the direction of time as in the reference you suggest) yields highly competitive if not superior results on common benchmark tasks. This has implications for the design of new benchmarks, among other things. Concerning scalability of our method, this concern was shared by a few other reviewers, and in response, we added a scalability subsection to the appendix. We also address this in detail in some other responses. As a short summary, we find that our method, exactly as presented in the paper, can be run on fairly large datasets (like the 800K+ edge Epinions) on a standard 16GB laptop, and we propose a few simple and straightforward extensions for greater scalability.

---

### Official Review · Reviewer_Vkhz · 2022-11-01

**Confidence:** 4
**Correctness:** 3
**Technical Novelty And Significance:** 2
**Empirical Novelty And Significance:** Not applicable
**Recommendation:** 3

**Clarity, Quality, Novelty And Reproducibility:**

See strength and weakness.


**Strength And Weaknesses:**

Strength:
- Well written paper. The paper states its core contributions clearly and directly aims to address related works limitations. As a result, the readers would have a relatively easy time to understand the concepts and proposals written in the article.
- Extensive experimental studies. The author supported their work with a large set of experiments centering around node classification on a variety of datasets. I appreciate the effort.
- Ablation study. I appreciate the model study conducted on synthetic dataset, and showcasing TDLG's ability to learn the global structure of the graph.
- Very fast runtime of this method compared to the related works.

Weakness:

- Graph structure's enormous scale. The original graph with O(NT) nodes, where N is number of nodes and T is number of time steps, would be constructed as O(N^2 T) nodes, which is a quadratic scale. As is shown in table 2, the #edges are typically much larger than the #nodes, thus rendering heavy memory pressure of the model runtime. However, this weakness is partially offset by its fast runtime. But this limitation makes the model hard to deploy on edge devices.

- Reliance of global graph structure. Based on section 2's experimental setup explanation (Temporal edge embedding and classification, Temporal link prediction), the model proposed in this paper relies on adjacency matrix A's row as its input feature to predict the edge class. This creates a limitation that the training and inference has to be operated on the same graph. Whereas, the test data coming from a different graph structure will make this method infeasible. My main concern is that the neural net training/testing on the same graph is not actually learning signatures of edges, and that how those edges are interacting with its important neighbors, but rather implicitly learning the training graph’s full topological information, thus making the inference on test set much easier, at the cost of poor generalizability on unseen graphs. Note, the models ability on learning the global graph structure is supported by the author's model study experiment in section 3,4, see figure 2.

- Reliance of global graph structure cond. However, I understand certain applications may actually favors this setup, i.e. recommender systems. As such, let us relax my argument by a little bit and assume we always operate on the same global graph. Then, if we have an unseen edge during test time, and we have no such row in the matrix A. In order to compute its feature, we have to rely on the set of training edges to acquire a relationship row vector between the test edge to all train edges. Our feature set is then limited/bounded by those training edges. However, the test edges can formulate interactions with edges that are not found in the training set, and we will not be able to acquire those features. This is a critical limitation, and this is also the core reason why cold start is always a tricky and heated topic in recommender systems.

- Reliance of global graph structure cond. That being said, I would be interested to see if the authors could compute localized embedding computation based on neural functions between nodes, and use them for edge classification tasks. For example, one could use graph attention networks [1], or spectral graph convolution kernels [2][3] to compute node embeddings on the line graph, thus constituting a valid representation of the temporal edge in the original graph. When this method is equipped with localized embeddings, we can train on a set of graphs and eval on unseen graph structures without issues.

- Experimental results are not completely fair. Most of the related work, for example evolve GCN is not depending on the global graph structure, thus being able to generalize to unseen graphs. When comparing to those kind of more generalizable work, this paper intrinsically has an advantage: it leverages the global graph structure and will not work if it is tested on an unseen graph. This is probably a good reason why this method is performing on 90% level across the board while the related works only performs at 70% level. In addition, figure 4 hints that when the access to global graph structure is impaired (when we only use 128 eigenvector's projections), the model's performance is severely hindered.


[1] Graph attention networks. https://arxiv.org/abs/1710.10903
[2] Spectral graph convnets. https://arxiv.org/abs/1606.09375
[3] Spectral localized kernels on point clouds. https://arxiv.org/abs/1803.05827


**Summary Of The Paper:**

This paper proposes to directly model feature embeddings from a line graph. The line graph is comprised of nodes as the original temporal edges, and new set of edges between the line graph nodes are captured by interactions between the original temporal edges.

The paper addresses critical ML application in the context of dynamic graph structures that is changing as time evolves, and how to conduct representation learning for such dynamic structures. The limitation of discretization of the time dimension is addressed and a great reduction in runtime compared to other state of the art methods are also demonstrated. The resultant method is simple yet effective, and resembles classical methods in recommendation systems which also adopts the weight adjacency matrix to derive embeddings.

**Summary Of The Review:**

At the present form of the paper, I am discouraged to recommended it with high ratings due to the limitation of its reliance on the full graph structure being consistent for training and inference time. Granted, there are applications scenarios that favors this setup, but it is indeed a limitation for generalizable representation learning.

The experiments and evaluation process are also compared against more generalizable methods which could operate on unseen graphs during test time, thus creating an unfair comparison. I would be more convinced if the compared methods are also operating on the same graph throughout and leveraging the global graph structure during its learning process.

---

> ### Author Response · Authors · 2022-11-18
> **Response to Reviewer Vkhz**
>
> Thank you for the review and feedback. We address the main comments, which concern scalability and “global graph structure.”
>
> ### Scalability
> Thank you for raising this important point, which is a common concern among reviewers. We have added some discussion of scalability in a new subsection of the appendix, and restate a bit as follows.\
> The number of nonzero elements in a line graph scales with the sum of squared degrees of the nodes. In theory, this could be very expensive, but real-world degree distributions generally seem to make this practical, at least up to the fairly large Epinions dataset of over 800K edges. We are able to run our method on Epinions on a standard 16GB laptop, which is not true of many of the comparison methods. We include a plot of the number of nonzero elements in the TDLG matrices of the real-world graphs, relative to the sizes of the embeddings of other methods, and find that they are generally comparable.\
> To scale our method to much larger datasets of millions of edges, there are several possible strategies, and we now discuss a few of these. First, as one reviewer suggested, some edges will share an endpoint but be very far apart in time. Hence the entry corresponding to such pairs of edges will be nonzero but very small; these entries could be ignored. Second, and perhaps most importantly, it is not necessary to hold the entire TDLG matrix in memory at once: given the closed-form formula, it is possible to construct some rows and supply them to a stochastic solver on-the-fly. This may be especially helpful in the edge device setting that you mention. Third, since each column can be thought of as a feature, it may be possible to sample only certain columns, e.g., by leverage score sampling, without much drop in performance. We do not employ any of these strategies in this work since they are not necessary for the datasets we consider.
>
>
>
> ### Global graph structure / Use of sparse feature vectors
>
> The quotes we would like to highlight concerning this comment are
>
> > This creates a limitation that the training and inference has to be operated on the same graph
>
> > at the cost of poor generalizability on unseen graphs
>
> Based on these quotes, we believe you are referring to the difference between inductive and transductive graph ML – thank you for raising this point. Our method is designed to work for the transductive setting, with edges arising from a single network. It would be interesting to explore if an approach similar to ours could also apply to the inductive setting (e.g., when the training and test data come from totally separate sets of networks), and you have noted some possible directions for extending a line graph method to this setting.
>
> We have now included this as a future direction in the conclusion, and make it clear that this is outside the scope of our paper.
>
>
>
>
> > The experiments and evaluation process are also compared against more generalizable methods which could operate on unseen graphs during test time
>
> With the sole exception of VGRNN, none of the papers for these compared methods seem to evaluate or even mention the inductive setting. We believe some of the other compared methods could perhaps generalize to inductive settings (like EvolveGCN), whereas others could not (like TIMERS). Upon reading this review, we did also find another paper (Xu et al.) which explores the inductive setting for temporal graphs – this seems like an interesting problem, even if it is not the focus of this work.
>
> > figure 4 hints that when the access to global graph structure is impaired (when we only use 128 eigenvector's projections), the model's performance is severely hindered
>
> As you state, this 'dense’ SVD version of our method is indeed less performant than the main 'sparse’ version, but even under the constraint of needing the network information to be compressed (i.e., to 128 dimensions, limiting access to the precise 'global graph structure’ and hence necessitating learning 'signatures’ of edges), our method still outperforms others on edge classification, as shown in the figure.
>
> ---
>
> Wu, Felix, et al. "Simplifying graph convolutional networks." International Conference on Machine Learning. PMLR, 2019.
>
> Xu, Da, et al. "Inductive representation learning on temporal graphs." International Conference on Learning Representations, 2020.

---

> > ### Comment · Reviewer_Vkhz · 2022-12-03
> > **Thanks for the reply!**
> >
> > I read through the reply and I think my original concerns are still hold valid. And thanks to the authors for making things clear in their newer version of the paper.
> >
> > My rating remains the same.

---

> > > ### Author Response · Authors · 2022-12-03
> > > **Response to Follow-up**
> > >
> > > Thank you for following up. We believe that the scalability concern is reasonably addressed above, and we also believe that our focus on the transductive setting should not be inherently disqualifying (assuming your criticism of "reliance on global graph structure" indeed referred to this), so we respectfully disagree with your conclusion. However, we are thankful for your time and consideration in reviewing our work.

---

### Official Review · Reviewer_HmSu · 2022-11-02

**Confidence:** 4
**Correctness:** 4
**Technical Novelty And Significance:** 3
**Empirical Novelty And Significance:** 3
**Recommendation:** 8

**Clarity, Quality, Novelty And Reproducibility:**

The proposed idea has some novelty. The paper is well-structured and well-written. The figures and the tables support the text and are well done. The authors provide a demo python notebook ready to use in the supplementary material for reproducibility. All datasets used are publicly available.

**Details Of Ethics Concerns:**

No ethical concerns

**Strength And Weaknesses:**

Strengths:
- The paper is well-structured and well-written. The paper is easy to follow and the flow is great.
- The authors have done a good job with the related works; they have organized most of the state of art methods and they discuss about the limitations and the challenges of each category which also gives motivation for the proposed model and setup.
- All figures and tables are well-thought and designed. The authors make the best out of the 9 page limitation.
- The demonstrating example and the theoretical discussion give the reader the extra proofs and background needed for this methodology.
- The results are very promising and show that the proposed method outperforms the six comparison models in five datasets in both edge classification and edge prediction tasks.
- The runtime results also show that the proposed method reduces the runtime, while increases the performance, and it also highlights that a simpler method can have comparable or better performance to other more complicated methods.

Weaknesses:
- The novelty of the work is incremental.
- The fact that it is proposed to create the line graph using the continuous stream of timestamps information as weights is not fully convincing on how this is continuous and not discrete time related information. Maybe more explanation and justification is needed in that part.
- It would be interesting to see all the models that appear in Table 1, in the comparison methods in the experimental setup.
- Do the train/test split percentages refer to the number of days? Can later days appear in the train set and earlier days in the test set? How do the authors do 10 trials with 10 random splits given that the time has the concept of order in it?

**Summary Of The Paper:**

This paper is about edge representation learning in continuous temporal networks. The proposed method (TDLG) starts with the construction of the line graph which converts each edge to a node and connects interactions that share the same endpoint node.Then, the edge weights in the line graph are calculated based on  the time differences between interactions, with interactions that happen closer in time to have higher weights. The line graph then represents both topological proximity and temporal proximity. Efficient classical methods can then be applied to the line graph. The authors of this paper propose the temporal stochastic block model (TSBM), which naturally extends the stochastic block model for static networks to temporal networks. The results contain a comparison of six state of the art models to the proposed TDGL on five real world datasets. The focus is on two downstream tasks: the temporal edge classification task and the {inter/extra}-polative link prediction task. The TDGL in most cases outperforms all other comparison methods. There is also discussion about the runtime improvement, the hyperparameters and in the appendix a full report on the results can be found.

**Summary Of The Review:**

Overall, this is a well-written paper, the quality of the work and the presentation is high. The proposed method is simple with incremental novelty, but it outperforms all other comparison state of the art models in five datasets and has a shorter runtime. The authors have done a great job describing the proposed method, and also in the experimental design and implementation.

---

> ### Author Response · Authors · 2022-11-18
> **Response to Reviewer HmSu**
>
> Thank you very much for the review. We respond to two points you raise.
>
> > The fact that it is proposed to create the line graph using the continuous stream of timestamps information as weights is not fully convincing on how this is continuous and not discrete time related information. Maybe more explanation and justification is needed in that part.
>
> To clarify, the temporal edges themselves are discrete objects. However, in the raw datasets, each of these edges is associated with a real-valued, 'continuous’ time. Most prior methods bin these continuous time into some small, finite number of buckets, e.g., 20. This produces a time-series of static (i.e., non-temporal) graphs that the method works with. By contrast, our method does not require this procedure, since it works with the real-valued times directly.
>
> >Do the train/test split percentages refer to the number of days? Can later days appear in the train set and earlier days in the test set? How do the authors do 10 trials with 10 random splits given that the time has the concept of order in it?
>
> For edge classification, we simply take all the edges, randomly split them into train/test, provide the training edges to the classifier, then test on the remainder (see first paragraph of Section 6.1). This is the usual setup, which is agnostic to time.
> Recall that we test two settings for link prediction. For both, the training data comprises some random 70% of all data excluding the 'late’ edges from the last time period. This random selection is where the difference in trials comes from. For the interpolative setting, the test data comprises the remaining 30%, again excluding the 'late’ edges. For the extrapolative setting, the test data comprises all of the 'late’ edges. This is described in Section 6.2.

---

### Official Review · Reviewer_CTmU · 2022-11-02

**Confidence:** 4
**Correctness:** 3
**Technical Novelty And Significance:** 3
**Empirical Novelty And Significance:** 3
**Recommendation:** 8

**Clarity, Quality, Novelty And Reproducibility:**

- The writing was clear, easy to follow and logical. I appreciated the explicit example given after exposition of their framework since it helped me gain a fast intuition about the problem.
- I'm not expertly familiar with the relevant citing literature, so am unsure about the novelty of the work in the context of the relevant literature. A major strength of this is the simplicity of the work, but it seems unlikely to me that it wouldn't have been considered before.
- The results are strong and convincing. I think this is a major strength of the work.
- The writing is very clear.
- I think the work is reproducible. However, I didn't see mention of an implementation repository which hinders fast and easy reproduction.
- I didn't find the theory too convincing mainly because of the strong assumptions that are associated with it. I am not certain about my position, but I didn't see the connection between the theory and the practical results either.

**Strength And Weaknesses:**

Pros:
+ Simple methodology that seems to make a lot of intuitive sense. I found the demonstrative example rather compelling given its simplicity and clear outcomes. I think I would have valued (for comparative purposes) examples of baselines
+ Experimental results look good, both in terms of accuracy and runtime. They seem to dominate the baselines considered.

Cons:
- The edge weight function, being an RBF kernel, is 'opinionated' as to the nature of the weighting function. I would imagine that a richer function class (even a NN) could allow much richer relationships to be learnt in a data driven manner. Is this a weakness? I would have liked abalation studies with other weight functions, such as those mentioned previously.
- I didn't see a strong connection between the theoretical analysis and evaluation. In particular, the reduction of the time variance to 0 in the analysis seems to me to implicitly discretise the context of the analysis, and its conclusions have (I believe) very weak links to the continuous time problem.
- I am curious about spectral interpretations of this construction, particularly regarding the specturm of the edge adjacency matrix.
- In hyperparam selection section, I would value clarification on what fraction of training data was used for valudation. The authors explicitly train and test sets, but not validation.

Additional questions:
- Why is escorts sensitive to the time scale hyperparameter and the others aren't?
- Why is it that normalisation hinders the classification performance?
- The method seems so simple that I'm surprised it hasn't been elucidated before.
- The structure of your adjacency matrix (Eq (1)) looks similar to the product of two kernels (linear kernel and an squared exponential kernel). I wonder if theory could be borrowed from the kernel and / or Gaussian processes literature to help with the understanding of the domain here?

**Summary Of The Paper:**

Edge embeddings and continuous time have not received as much attention as the alternatives between these two categories, and this paper takes an edge-first approach to temporal graph edge embeddings. The paper introduces a simple strategy to obtain these, justifying several useful aspects of their strategy. A demonstrative example is given to share intuition with the readers, which is followed by theoretical insights on the methodolgy. Following the related work section, empirical analysis is performend, where the proposed method beats several baseline methods. Given that this paper is focusing on a niche, I'm not sure if any method can be considered SoTA. Empirically, the proposed method ran to conlusion over all datasets, whereas some baseline methods did not.

**Summary Of The Review:**

This paper introduces a straightforward approach to edge embeddings of temporal graphs. The proposed approach works very well according to the empirical results. I also find the similicity of the approach to be a strength rather than a weakness of the paper, though, not being an expert in this area, I would be surprised if this didn't exist elsewhere. I appreciated the theoretical analysis, but I am wary of some aspects, particularly that the assumptions (esp. the zero time variance) means the analysis is more suitable discrete rather than continuous contexts, which is not the focus of this paper. If other reviews, or the authors, can ratify the theory and novelty of the contribution I would be willing to increase my recommendation.

---

> ### Author Response · Authors · 2022-11-18
> **Response to Reviewer CTmU Part 1**
>
> Thank you for the extensive review and insightful suggestions. As you recommend, we try to ratify the theory and novelty of our contributions through the following response.
>
> > The edge weight function, being an RBF kernel, is 'opinionated' as to the nature of the weighting function… Is this a weakness?
>
> > The method seems so simple that I'm surprised it hasn't been elucidated before.
>
> Indeed, the general concept of TDLGs is not fixed to an RBF kernel – as we note in the paper, rather than Gaussian decay, one could just as well consider Laplacian decay, and a more exotic kernel may be appropriate in some cases, e.g., some sinusoidal kernel for periodic data. We purposely focus on the simplest, most natural kernel to emphasize our main point, that a very simple method, with the basic inductive prior of 'locality’, can be highly effective on the common benchmarks of this area. This says as much about the common benchmarks/datasets as it does about the prior, more complex methods. In particular, our results suggest that a method going beyond the capture of edges’ temporal locality (in our method, from the RBF kernel) and topological locality (in our method, from the line graph) is 'overkill’ on these datasets, at least as concerns achieving our method’s performance. We note that there is recent precedent for a simple method performing favorably to prior deep methods – see Wu et al. (2019).
>
> > I didn't see a strong connection between the theoretical analysis and evaluation. In particular, the reduction of the time variance to 0 in the analysis seems to me to implicitly discretise the context of the analysis, and its conclusions have (I believe) very weak links to the continuous time problem.
>
> The theoretical analysis mainly serves as a sanity check on our method’s capabilities. It confirms that our method can leverage time information to derive network structure, even in especially difficult situations where it is otherwise impossible. Specifically, this is referring to the motivating synthetic network in Figure 1: even if the time distributions for the two SBMs are shrunk to constant values, distinguishing the communities requires using this time information, and we prove that our method is capable of this.\
> The removal of time variance greatly simplifies the analysis, but it is not entirely divorced from a continuous-time context. In general, the proposed TSBM is meant to concretize the context of interactions between two communities of nodes, where the ratio of in-group to out-group interactions varies over time. The theoretical result works towards showing that our approach, and perhaps more generally, purely locality-based approaches, can be useful for this context. A further, second step might be applying matrix concentration bounds to show that the result also holds with small but non-zero time variance. We agree that our current theoretical analysis does not separate the capabilities of our method from methods that discretize time. Further theoretical analysis that does so would be very interesting – we expand on possible routes for this analysis in our response to Reviewer GdB7.
>
> We also note that our method is fairly unique in allowing for such an explicit analysis, even in a synthetic scenario, and this is thanks to its simplicity.

---

> > ### Author Response · Authors · 2022-11-18
> > **Response to Reviewer CTmU Part 2**
> >
> > > I am curious about spectral interpretations of this construction, particularly regarding the specturm of the edge adjacency matrix.
> >
> > > The structure of your adjacency matrix (Eq (1)) looks similar to the product of two kernels (linear kernel and an squared exponential kernel). I wonder if theory could be borrowed from the kernel and / or Gaussian processes literature to help with the understanding of the domain here?
> >
> > These are very interesting questions. While there is some prior work on the spectrum of line graph adjacency matrices (see, e.g., Van Dam et al. 2003), the application of nonlinear RBF/Gaussian decay to the entries will complicate the spectrum. It is a really insightful observation that the TDLG adjacency matrix is the Hadamard product of two kernel matrices. This implies some interesting things, e.g., that the matrix is PSD, along with some basic eigenvalue bounds using the eigenvalues of the unweighted line graph adjacency matrix and the gaussian kernel matrix. These could be useful in extending our theoretical results.
> >
> > >In hyperparam selection section, I would value clarification on what fraction of training data was used for valudation. The authors explicitly train and test sets, but not validation.
> >
> > We only informally tune our hyperparameters, so we do not form validation sets. We have added some details of this tuning process in Appendix A.3. We note that our method has a single hyperparameter, $\sigma_t$, and the value $10^{-1}$ seems to work well across all the datasets. For the comparison methods, we use default hyperparameters from the implementation we referenced (Liu et al. 2020).
> >
> > > Why is escorts sensitive to the time scale hyperparameter and the others aren't? Why is it that normalisation hinders the classification performance?
> >
> > These are interesting questions. We did look into the variance of times in Escorts to see if this would explain its unique behavior, but the variance is in line with the other datasets. Loosely, we speculate that the reason is that the correlation between temporal edges / interactions (in the case of this dataset, sex buyers rating sex sellers) happens at a very specific time scale, as opposed to other datasets where this correlation is more diffuse. As for normalization hindering the classification performance, again we only speculate: we believe the reason is that normalization interacts adversely with the regularization of the logistic regression classifier, at least in the ways that we implement the two.
> >
> > > I think the work is reproducible. However, I didn't see mention of an implementation repository which hinders fast and easy reproduction.
> >
> > We have included a full demo of our method in the supplementary material, and we would upload this to a public repository upon acceptance.
> >
> > ---
> >
> > Wu, Felix, et al. "Simplifying graph convolutional networks." International Conference on Machine Learning, 2019.
> >
> > Van Dam, Edwin R., and Willem H. Haemers. "Which graphs are determined by their spectrum?." Linear Algebra and its applications 373 (2003): 241-272.

---

> > > ### Comment · Reviewer_CTmU · 2022-11-28
> > > **Response**
> > >
> > > Thank you for your detailed response to my questions.
> > >
> > > I'm in two minds about changing my score. On the one hand, the method is very simple and useful (two strong positives), but on the other, I'm still unconvinced about the theory. I appreciate that relaxing the constraint on zero variance is non-trivial, but I feel that this could also be considered as a smoke and dagger approach to discretise a continuous problem, and as a result I'm still unconvinced that it has much value to this particular domain.
> > >
> > > To break the tie, I have read all of the reviews and responses in detail. I agree that the response to GdB7 adds useful context, and I am inclined to increase my score as a result. Will this sketch be incorporated into the main paper?
> > >
> > > Nitpick: Can you explain please what is meant by "informally tuning" parameters? If this is a well known term, it has passed my notice. My understanding is that they are tuned on the test set - is this correct?

---

> > > > ### Author Response · Authors · 2022-11-30
> > > > **Response to Follow-up**
> > > >
> > > > Thank you for following up. We really appreciate the effort to carefully read the other reviews and responses, and we are glad the response to Reviewer GdB7 adds some context to the theoretical analysis. As we note there, we believe that our theoretical analysis does not conceptually rely on trivial solutions that are possible in the regime of discrete / zero variance times. We will be sure to incorporate the sketch we outline there, for extension to the non-zero variance setting, into the paper.
> > > >
> > > > > "informally tuning" hyperparameters... they are tuned on the test set - is this correct?
> > > >
> > > > Yes: In all experiments, we report average performance over several random train/test splits (no validation). In initial experiments over different variants of our method, we found $\sigma_t = \tfrac{1}{10}$ to work well (over other such random splits), and we fixed that value for all further experiments. It would have been more proper to hold out some test data the whole time, since now $\sigma_t$ has been loosely "fit" to the test data, but we believe this has minimal impact since: 1) there is only this one hyperparameter, 2) values around $\sigma_t = \tfrac{1}{10}$ seem to work well across datasets (see Figure 6), and 3) performance is fairly robust to minor perturbations of $\sigma_t$ on most of the datasets we test.

---

### Official Review · Reviewer_Pn6V · 2022-11-03

**Confidence:** 4
**Correctness:** 3
**Technical Novelty And Significance:** 2
**Empirical Novelty And Significance:** 3
**Recommendation:** 6

**Clarity, Quality, Novelty And Reproducibility:**

Clarity:

- This paper is clearly written and easy to read.

Quality:

- This paper is well-organized.

Novelty:

- Despite the idea to utilize the row vectors of the line graphs with time-decayed weight being straightforward, there has been no work that shows such a simple method surprisingly works well.

Reproducibility:

- Readers can easily implement the proposed method.


**Strength And Weaknesses:**

Strengths:

- The idea to utilize line graphs in data mining tasks has been discussed in a lot of papers such as [1], and it seems straightforward to weigh the line graph according to time decay between edges; however, to my knowledge, there has been no work that shows such simple method can surprisingly outperform other methods that discretize the time stamps.
- This paper is well-organized and clearly written. It is easy for readers to follow.

Weaknesses:

- The discussion and evaluation related to the time/space complexity are not provided enough. Because the embedding vectors have fundamentally extremely high dimensionality, the computation of the downstream tasks will be affected by the sparsity of the vectors, which depends on the size and fraction of hub nodes in the original graph. Moreover, the computation cost of eigendecomposition should be discussed more including how to update the eigendecomposition in the time-evolving graphs.

[1] Ahn, YY., Bagrow, J. & Lehmann, S. Link communities reveal multiscale complexity in networks. Nature 466, 761-764 (2010).


**Summary Of The Paper:**

This paper discusses how to embed the edges of temporal networks. It proposes to create line graphs weighed by Gaussian weight decay and to use the row vectors of the adjacency matrix of the line graph as the embeddings of the edges. The experimental results show that the proposed method performs better in edge classification tasks and link prediction tasks than other methods that discretize the time stamps of the datasets.

**Summary Of The Review:**

This paper is well-organized and clearly written. Despite the idea to utilize the row vectors of the line graphs with time-decayed weight being straightforward, there has been no work that shows such a simple method surprisingly works well. A downside is that this paper does not provide enough discussion and evaluation of time/space complexity.

---

> ### Author Response · Authors · 2022-11-18
> **Response to Reviewer Pn6V**
>
> Thank you for the insightful suggestions. We have now included a new subsection in the appendix discussing scalability and the time/space complexity of the approach.
>
> ### Scalability
> Thank you for raising this important point. We restate a bit of the new subsection as follows.
> The number of nonzero elements in a line graph scales with the sum of squared degrees of the nodes. In theory, this could be very expensive, but real-world degree distributions generally seem to make this practical, at least up to the fairly large Epinions dataset of over 800K edges. We are able to run our method on Epinions on a standard 16GB laptop, which is not true of many of the comparison methods. We include a plot of the number of nonzero elements in the TDLG matrices of the real-world graphs, relative to the sizes of the embeddings of other methods, and find that they are generally comparable.\
> To scale our method to much larger datasets of millions of edges, there are several possible strategies, and we now discuss a few of these. First, as one reviewer suggested, some edges will share an endpoint but be very far apart in time. Hence the entry corresponding to such pairs of edges will be nonzero but very small; these entries could be ignored. Second, and perhaps most importantly, it is not necessary to hold the entire TDLG matrix in memory at once: given the closed-form formula, it is possible to construct some rows and supply them to a stochastic solver on-the-fly. Third, since each column can be thought of as a feature, it may be possible to sample only certain columns, e.g., by leverage score sampling, without much drop in performance. We do not employ any of these strategies in this work since they are not necessary for the datasets we consider.
>
> > Because the embedding vectors have fundamentally extremely high dimensionality, the computation of the downstream tasks will be affected by the sparsity of the vectors, which depends on the size and fraction of hub nodes in the original graph.
>
> Indeed, the logistic regression takes longer with our main method's long, sparse embedding vectors than with small, dense ones. However, the values in the runtime plot include both the computation of the embeddings and the downstream task (i.e., training/inference with the logistic regression classifier using the embeddings), and we generally find that our method achieves the lowest total runtime on the tested datasets.
>
> > the computation cost of eigendecomposition should be discussed more including how to update the eigendecomposition in the time-evolving graphs.
>
> In this work, we generally consider the situation where the input is a graph with timestamped edges, but not a streaming setting where the graph further changes after processing. Extending our method to this situation is an interesting direction. With iterative methods for truncated SVD, it should be possible to to update the embeddings efficiently as the underlying matrix changes: When an edge is added, this just adds a row/column to this line graph adjacency matrix. Hence it is a low-rank update of the adjacency matrix, so the truncated SVD, and thus the embeddings, can be updated without fully recomputing it (see, e.g., Brand 2006).\
> We would also like to emphasize that our main method uses sparse feature vectors, without any eigendecomposition. The `dense’ variant of our method, which does involve eigendecomposition, is mainly included since it is, in some sense, a more direct comparison with other methods, which also use dense embeddings. Generally, with $m$ nonzero entries, the cost of an iteration of truncated SVD where we target the top $k$ singular vectors (e.g., for $128$-dimensional embeddings) is roughly $O(mk)$. As discussed above, the line graph adjacency matrix is typically very sparse, so this is much more efficient than SVD of an arbitrary $m \times m$ matrix. The new scalability section of the appendix includes some discussion of this.
>
> ---
>
> Brand, Matthew. "Fast low-rank modifications of the thin singular value decomposition." Linear algebra and its applications 415.1 (2006): 20-30.

---

### Official Review · Reviewer_C63A · 2022-11-03

**Confidence:** 2
**Correctness:** 3
**Technical Novelty And Significance:** 3
**Empirical Novelty And Significance:** 3
**Recommendation:** 5

**Clarity, Quality, Novelty And Reproducibility:**

The paper introduces the problem clearly and has clear contextual structure.

The motivation of why edge embedding needs to be improved is unclear to me. It is stated that edge embedding has seen less interest, while it does not make a strong and effective motivation.


Is $\lambda$ in Proposition 4.1 a typo? Should be $\gamma$?

The time scale hyperparameter $\sigma_{t}$ is important and is chosen by informal tuning. Can you explain informal tuning?


The proposed approach provides a novel way of modeling edges in temporal networks.

The related work and baseline approaches are papers proposed in year 2019 or before. It would be better to include more recent papers to have a better reflection on the novelty of this paper.


The reproducibility of this paper is good since a code demo is provided.


**Strength And Weaknesses:**

Strength:

Overall the paper has good clarity in the problem statement. The novelty is stated clearly.

Weakness:

W1: The motivation of why edge embedding needs to be improved is unclear to me. It is stated that edge embedding has seen less interest, while it does not make a strong and effective motivation.

W2: The related work and baseline approaches are papers proposed in year 2019 or before. It would be better to include more recent papers to have a better reflection on the novelty of this paper.


**Summary Of The Paper:**

This paper aims to directly model the edges in temporal networks instead of indirectly inferring edge embeddings by computations from nodes. To achieve so, the paper constructs Time-Decayed Line Graphs (TDLGs) to use each node to represent the edges, and weigh the edges between resultant nodes based on differences in time. Such modeling avoids the assumption of discretized time info. Theoretical analysis is provided to ensure the proposed edge modeling can recover original nodes based on the stochastic block model (SBM).


**Summary Of The Review:**

Overall, the paper is well written and the contents are self-contained. The motivation and the lack of more recent related work are the two main issues.

---

> ### Author Response · Authors · 2022-11-18
> **Response to Reviewer C63A**
>
> Thank you for the review. We address some minor points, then the broader comments:
>
> >Is $\lambda$  in Proposition 4.1 a typo? Should be $\gamma$?
>
> Yes, thank you for pointing out this typo. We fixed it.
>
> > The time scale hyperparameter $\sigma_t$ is important and is chosen by informal tuning. Can you explain informal tuning?
>
> In initial experiments on small graphs, we tried some natural values: $10^{-2}$, $10^{-1}$, and $10^0$. We found $10^{-1}$ to appear to work well in general, and all subsequent experiments on the 5 datasets of the paper, which appear in the paper’s main body, use this value. We later conduct experiments on these datasets as $\sigma_t$ is varied (see Appendix A.3) which confirm that this value generally performs well. We have added this information to the paper.
>
> ### Need for edge embeddings
> As suggested, we have now clarified this in the introduction. Please see the last 3 sentences in paragraph 2 and paragraph 3 of the introduction. Most importantly, temporal networks arrive in the form of an edge stream where each edge is timestamped; therefore, it is only natural to directly derive embeddings for the edges, as opposed to the nodes. It is also practically important since all of the downstream tasks that we are interested in, such as edge classification, require edge embeddings. It is less natural to learn embeddings for nodes, which do not actually have timestamps on them, and then use these embeddings to compute edge embeddings that can be used for the downstream applications.
>
>
> ### On including more recent papers
> Thank you for raising this point. We have now included a few more recent papers in the related work as suggested: TGAT (Xu et al. 2020), TGNs (Rossi et al. 2020), MeTa (Wang et al. 2021), and PINT (Souza et al. 2022).
>
> Xu, Da, et al. "Inductive representation learning on temporal graphs." International Conference on Learning Representations, 2020.
>
> Rossi, Emanuele, et al. "Temporal graph networks for deep learning on dynamic graphs." International Conference on Learning Representations, 2020.
>
> Wang, Yiwei, et al. "Adaptive data augmentation on temporal graphs." Advances in Neural Information Processing Systems, 2021.
>
> Souza, Amauri H., et al. "Provably expressive temporal graph networks." arXiv preprint arXiv:2209.15059 (2022).

---

### Decision · Program_Chairs · 2023-01-20

**Decision:**

Accept: poster

**Justification For Why Not Higher Score:**

The problem studied is a bit niche and it may not be of interest for a wider audience. The theoretical results are nice but not particularly strong or surprising.

**Justification For Why Not Lower Score:**

The paper study a natural problem and introduce a novel idea. The paper contains both nice theoretical and empirical results.

**Metareview: Summary, Strengths And Weaknesses:**

The authors propose a new model for temporal network. The key idea behind the paper is to design a linearization of the evolution of the graph. In this way they can avoid standard shortcoming of previous techniques. In particular, they introduce a Time-Decayed Line Graphs (TDLGs) representation that can be used to weigh the edges between nodes based on differences in time.

The proposed model is simple but it has good experimental performances. In addition, the authors also provide theoretical results showing some nice properties of their technique.

Overall, the paper would be an interesting contribution to the ICLR program.

**Note From Pc:**

if the above contains the word "oral" or "spotlight" please see: "oral" presentation means -> notable-top-5% and "spotlight" means -> notable-top-25%. As stated in our emails, we are disassociating presentation type from AC recommendations